# Respiratory viruses Ct values and association with clinical outcomes among adults visiting the ED with lower respiratory tract infections

Ines Bentahar[1], Paul Loubet[2,3], Florian Salipante[4], Christophe Choquet[1,5], Diane Descamps[5,6], Benoit Visseaux[7], Nathan Peiffer Smadja[5,8], Quentin Le Hingrat[5,6], Donia Bouzid [1,5]*

1 AP-HP Nord, Emergency Department, Bichat-Claude Bernard University Hospital, Paris, France, 2 Department of Infectious and Tropical Diseases, CHU Nîmes. Université Montpellier, Nîmes, France, 3 Université de Montpellier, VBMI, Inserm, Nîmes, France, 4 Department of Biostatistics, Epidemiology, Public Health and Innovation in Methodology, CHU de Nîmes, 5 Université Paris Cité, INSERM, IAME, Paris, France, 6 AP-HP Nord, Virology Department, Bichat-Claude Bernard University Hospital, Paris, France, 7 Infectiology Department, Cerba Laboratory, Cerba Healthcare, Frépillon, France, 8 Department of Infectious and Tropical Diseases, Bichat-Claude Bernard University Hospital, Paris, France

* donia.bouzid@aphp.fr

## Abstract

### Background

The correlation between real-time PCR (rt-PCR) cycle threshold (Ct) values for respiratory viruses and clinical outcomes remains unclear. This study evaluates the association between Ct values and clinical outcomes in patients tested via point-of-care testing upon emergency department (ED) admission.

### Methods

This is a retrospective analysis of adults admitted to a French university hospital ED for suspected lower respiratory tract infections (LRTI) requiring oxygen therapy between 2019 and 2020. Ct values were assessed for their association with symptom duration and clinical outcomes (hospital length of stay (LOS), Intensive Care Unit (ICU) admission, and 28-day mortality) using zero-inflated negative binomial regression (ZINB) and logistic regression models, adjusted for age, sex, co-infection, and symptom duration.

### Results

A total of 410 patients were included, with 37 (9%) having co-infections with two pathogens and 2 (0.5%) with three pathogens. The most common pathogens were human rhinovirus/enterovirus (HRV/EV) (26.3%), influenza A (24.9%), and SARS-CoV-2 (21.9%). Median symptom duration was 3 days [IQR: 2–7]. Of the patients, 308 (75.1%) were hospitalized, 74 (18%) required ICU care, and the 28-day mortality

**Data availability statement:** Data contain potentially identifying or sensitive patient information, and restrictions were imposed by the ethics committee. As a result, data cannot be made publicly available. However, data can be shared upon reasonable request and after submitting a formal application to the ethics committee. The restricting institution is Assistance Publique – Hôpitaux de Paris (AP-HP). Researchers who wish to request access to the data may contact the ethics committee via Naima Beldjoudi (naima.beldjoudi@aphp.fr), Assistance Publique – Hôpitaux de Paris (AP-HP).

**Funding:** This study is funded by Qiagen The funders had no role in study design, data collection and analysis, decision to publish, or preparation of the manuscript.

**Competing interests:** DB contributed to a funded symposium for Qiagen PL contributed to a funded symposium for Astra Zeneca This does not alter our adherence to PLOS ONE policies on sharing data and materials.

rate was 11.7% (n = 48). Multivariable analysis showed that higher Ct values for SARS-CoV-2 were associated with reduced odds of hospitalization (OR = 0.75, p = 0.04) and shorter LOS (x0.96 days per Ct unit increase, p = 0.04). Similar trends for shorter LOS were observed for HRV/EV and RSV but did not reach statistical significance. Conversely, higher influenza A Ct values were linked to longer LOS (x1.05 days per Ct unit increase, p = 0.025). Higher Ct values for SARS-CoV-2 were also associated with lower 28-day mortality (OR = 0.87, p = 0.049). Ct values were not associated with ICU admission for any virus.

## Conclusion

This study supports the association of higher Ct values with shorter LOS and lower mortality for SARS-CoV-2. In contrast, higher Ct values for influenza A were linked to longer LOS. Ct values were not predictive of ICU admission, underscoring the complexity of the relationship between viral load and clinical outcomes.

## Introduction

Lower respiratory tract infections (LRTI) remain one of the leading causes of mortality from infectious diseases worldwide [1]. Common viral causes of LRTI are influenza viruses, respiratory syncytial virus (RSV), parainfluenza virus (PiV), metapneumovirus (HMPV), adenovirus (HAdV), rhinoviruses (HRV), bocaviruses (HBoV), and coronaviruses (HCoV). Most of these viruses demonstrate seasonal circulation patterns, particularly in temperate regions [2,3]. These patterns have drastically changed since the emergence of SARS-CoV-2 at the end of 2019 [4,5].

Real-time PCR is increasingly used to detect and identify respiratory pathogens. The PCR cycle threshold (Ct) value represents the number of amplification cycles required to detect each tested pathogen. It is inversely proportional to the number of copies of the targeted genome in the sample, meaning that lower Ct values reflects higher pathogen loads. However, using Ct values as a proxy of the genomic load is influenced by various factors and may not be as reliable as a proper quantification. This data is often not disclosed with the test result [6,7]. Ct values have been widely discussed during the COVID-19 pandemic, and most studies agreed on the association between SARS-CoV-2 Ct values and hospital length of stay, mortality, and ICU admission [8,9]. However, such correlations for other respiratory viruses are still unclear to date. A recent systematic review has reported conflicting conclusions on this issue [10]. Some trends were consistent for RSV, with several studies reporting associations between low Ct values and clinically relevant outcomes such as ICU admission or hospital length of stay. For influenza viruses, the association between Ct values and hospitalization or ICU admission varied between studies. Nevertheless, the most extensive influenza study, conducted by Spencer et al., notably revealed that individuals with lower Ct values were more inclined to report moderate to severe disease symptoms and fever [11]. A parallel observation was noted in a

study of influenza A viral load where patients with pneumonia had more viral copies per milliliter than those with upper respiratory tract infections [12].

The objective of this study was to assess the association between respiratory viruses' Ct values at diagnosis and clinical outcomes in adults with suspected lower respiratory tract infections (LRTI).

## Methods

### Study design and participants

This study retrospectively reviewed electronic medical records of adult patients (≥18 years old) who presented to the emergency department (ED) of Bichat-Claude Bernard University Hospital (Paris, France) between January 1, 2019, and December 31, 2020. Patients were included if they exhibited symptoms of lower respiratory tract infection (LRTI), defined as at least one respiratory symptom (e.g., dyspnea, cough, expectoration) and one general symptom (e.g., fever, myalgia, headache), and required oxygen therapy at the time of admission. Respiratory pathogen testing was performed using multiplex PCR point-of-care testing for patients meeting these criteria.

A total of 410 patients with positive multiplex PCR results were included in the analysis. Demographic and clinical characteristics, including age, sex, symptom duration, comorbidities, and clinical outcomes (e.g., hospitalization, ICU admission, and mortality at day 28), were extracted from the medical records.

### Procedures

A nasopharyngeal swab was collected and tested using using the QIAstat-Dx Respiratory Panel V2 (Qiagen. Hilden. Germany). This rapid multiplex PCR assay detects 22 viral and bacterial respiratory targets, including influenza (A and B) viruses; parainfluenza viruses 1–4 (PIV); rhinovirus/enteroviruses (HRV/EV); respiratory syncytial viruses (RSV); human metapneumovirus (hMPV); adenoviruses (ADV); endemic coronaviruses (HKU1, OC43, NL63 and 229E); bocavirus; *Mycoplasma pneumoniae*; *Legionella pneumophila*; and *Bordetella pertussis*. Results were available within one hour. Ct values are available in the instrument and exported to the laboratory software.

### Statistical analysis

Semiquantitative data were expressed as median and interquartile range (IQR). Qualitative data were expressed as absolute number and frequency (%). Fisher exact tests were used in order to test for independence between qualitative variables.

The association between Ct values and Intensive Care Unit (ICU) admission and mortality at day 28 was assessed with logistic regression model. The association between Ct values and hospital length of stay (LOS) was assessed with a zero-inflated negative binomial regression (ZINB), only the tables of negative binomial regression model component are displayed in the results section. Both approaches were evaluated without and with adjustment on age, sex, coinfection, and symptoms duration.

The Ct values of the study patients were arbitrarily categorized as high (Ct 31–40), moderate (21–30) or low (11–20) [10].

Certain respiratory viruses, including adenovirus, HKU1, OC43, and metapneumovirus, were excluded from the detailed statistical analysis due to the limited number of cases detected during the study period. The small sample sizes for these pathogens restricted the statistical power to perform robust analyses and draw meaningful conclusions. As a result, only viruses with sufficient representation in the dataset were included in the final analysis to ensure reliable and interpretable results.

No imputation strategies were used to deal with missing data, a complete case analysis was used in the final dataset. Alpha risk was set to 5% in order to assess statistical significance.

The statistical analysis was performed with R version 4.2.0 (R Core Team (2020). R: A language and environment for statistical computing. R Foundation for Statistical Computing. Vienna. Austria. URL https://www.R-project.org/).

### Ethics

The study was approved by the CHU de Nîmes ethics committee (IRB number: 23.02.05).

All data were fully anonymized before the investigators accessed them on January 4th 2024, and ethics committee waived the requirement for informed consent.

## Results

### Baseline characteristics

Four hundred ten patients (36.3%) with a positive multiplex PCR were included in this analysis, with a median age of 68 years (IQR: 55–80). A slight majority were male (55.4%, n = 227). The median duration of symptoms was 3 days (IQR: 1–7), and 308 patients (75.1%) required hospitalization for at least one day, with a median hospital length of stay of 5 days (IQR: 1–11).

Baseline physiological and clinical measurements revealed a median pulse oximetry of 94% (IQR: 89–97%), a median respiratory rate of 20 breaths per minute (IQR: 17–28), and a median heart rate of 96 beats per minute (IQR: 83–112). Blood pressure values were also consistent with the patient cohort's clinical profile, with a median systolic pressure of 134 mmHg (IQR: 117–150) and diastolic pressure of 78 mmHg (IQR: 68–87). Glasgow scores indicated preserved consciousness, with a median score of 15 across the cohort.

The most frequently detected pathogens were rhinoviruses/enteroviruses (26.3%, n = 108), influenza A viruses (24.9%, n = 102), and SARS-CoV-2 (21.2%, n = 87). Co-infections were identified in 37 patients (9%) with two pathogens and 2 patients (0.5%) with three pathogens. Median Ct values varied by pathogen, with influenza A showing a median of 23 (IQR: 21–30), rhinoviruses/enteroviruses 26 (IQR: 23.5–31), and SARS-CoV-2 27 (IQR: 22.4–31). Less frequently detected pathogens, including metapneumovirus, parainfluenza, and RSV, had median Ct values ranging from 21 to 31, depending on the pathogen (Table 1).

### Ct value association with hospital length of stay

Regarding SARS-CoV-2, the ZINB model indicates that, after taking into account potential confounding factors such as age, sex, coinfection, and symptoms' delay, a high Ct value was associated with a decreased hospitalization rate (adjusted incidence rate ratio aIRR = 0.96 CI95% [0.92–0.99]. p = 0.04).

In the population of hospitalized patients (n = 308), higher Ct values for SARS-CoV-2 were associated with a shorter length of stay (x0.96 days for an increase of one unit of Ct, p = 0.04). Meanwhile, in the same population of hospitalized patients, higher Ct values for influenza A virus were associated with a longer length of stay in the hospital (x1.05 days for an increase of one unit of Ct, p = 0.025) (Table 2 and Fig 1).

The analysis did not find any association between the length of stay and high Ct values for RSV and HRV/EV (HRV/EV: aIRR = 0.98. CI95% [0.92–1.04], p = 0.53; RSV: aIRR = 0.97 CI95% [0.95–1]. p = 0.93) (Table 2 and Fig 1).

### Ct value association with ICU admission

The analysis showed no significant association between PCR cycle threshold (Ct) values and ICU admission for any of the pathogens studied. For all pathogens combined, the adjusted odds ratio (aOR) was 1.04 (95% CI: 0.99–1.09, p = 0.11), indicating that Ct values were not predictive of ICU admission. Similarly, individual analyses for specific pathogens, including SARS-CoV-2 (aOR: 1.00, 95% CI: 0.90–1.10, p = 0.99), influenza A (aOR: 1.02, 95% CI: 0.93–1.12, p = 0.61), rhinoviruses/enteroviruses (aOR: 1.02, 95% CI: 0.92–1.14, p = 0.65), and RSV (aOR: 1.05, 95% CI: 0.83–1.41, p = 0.69), revealed no significant associations. A detailed summary of these findings is provided in (Table 3).

**Table 1. Baseline demographic and clinical characteristics of the study population.**

|  | N | Median | Q1 | Q3 |
|---|---|---|---|---|
| Age (years) | 410 | 68 | 55 | 80 |
| Duration of symptoms (days) | 319 | 3 | 1 | 7 |
| Pulse oxymetry (%) | 410 | 94 | 89 | 97 |
| Respiratory rate (/min) | 410 | 20 | 17 | 28 |
| Systolic blood pressure (mmHg) | 410 | 134 | 117 | 150 |
| Diastolic blood pressure (mmHg) | 410 | 78 | 68.0 | 87 |
| Heart rate (bpm) | 410 | 96 | 83 | 112 |
| Glasgow score | 410 | 15 | 15 | 15 |
| Ct adenovirus | 6 | 33 | 32 | 34 |
| Ct coronavirus HKU1 | 16 | 25 | 22 | 31 |
| Ct coronavirus OC43 | 9 | 27 | 24 | 29 |
| Ct metapneumovirus | 28 | 21 | 16.7 | 26 |
| Ct rhinovirus/enterovirus | 108 | 26 | 23.5 | 31 |
| Ct influenza A | 98 | 23 | 21 | 30 |
| Ct influenza B | 12 | 20 | 18.5 | 22 |
| Ct parainfluenza | 14 | 31 | 29.2 | 34 |
| Ct RSV | 48 | 22 | 17.4 | 30 |
| Ct SARS-CoV-2 | 87 | 27 | 22.4 | 31 |

**Table 2. Association between PCR cycle threshold (Ct) values and hospital length of stay: univariate and multivariate analysis adjusted on age, sex, coinfections, and symptoms delays.**

|  | Univariate | | | | Multivariate | | | |
|---|---|---|---|---|---|---|---|---|
|  | IRR[*] | 2.5% | 97.5% | p.val | AIRR | 2.5% | 97.5% | p.val |
| All pathogens | 1.02 | 0.99 | 1.04 | 0.14 | 1.02 | 0.99 | 1.04 | 0.16 |
| Influenza A | 1.06 | 1.01 | 1.1 | 0.013 | 1.05 | 1.01 | 1.1 | **0.02** |
| SARS-CoV-2 | 0.96 | 0.92 | 1.01 | 0.09 | 0.96 | 0.92 | 1 | **0.04** |
| Rhinovirus/enterovirus | 0.96 | 0.9 | 1.02 | 0.16 | 0.98 | 0.92 | 1.04 | 0.53 |
| Metapneumovirus | 1 | 0.94 | 1.06 | 0.97 | 1.05 | 0.99 | 1.11 | 0.92 |
| RSV | 0.99 | 0.96 | 1.02 | 0.37 | 0.97 | 0.95 | 1 | 0.93 |

[*]**IRR: incidence rate ratio**

## Association of Ct value with mortality at day 28

After adjusting for age, sex, coinfections, and symptom duration, higher PCR cycle threshold (Ct) values for SARS-CoV-2 were significantly associated with decreased mortality at day 28 (adjusted odds ratio [aOR]: 0.87, 95% CI: 0.76–0.99, p = 0.04). This suggests that lower viral loads, as indicated by higher Ct values, are linked to improved survival outcomes for SARS-CoV-2 infections.

For other pathogens, no significant associations between Ct values and 28-day mortality were observed. The aORs were 0.96 (95% CI: 0.80–1.12, p = 0.59) for influenza A, 0.96 (95% CI: 0.83–1.11, p = 0.62) for rhinoviruses/enteroviruses, 1.03 (95% CI: 0.80–1.32, p = 0.78) for metapneumovirus, and 1.12 (95% CI: 0.95–1.36, p = 0.19) for RSV. (Fig 2 and Table 4).

No association, nor trends were identified for the other pathogens (S1 Table: Symptoms, Clinical patterns and outcomes according to the pathogen).

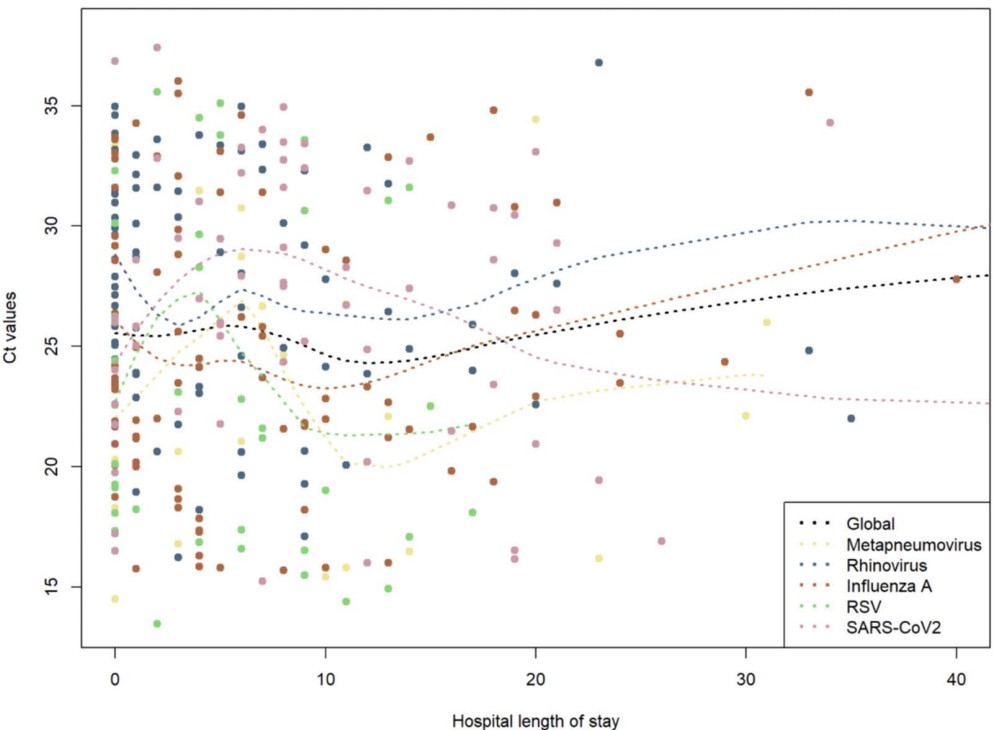

**Fig 1. Association between PCR cycle threshold (Ct) values and hospital length of stay.**

**Table 3. Association between PCR cycle threshold (Ct) values and ICU admission: univariate and multivariate analysis adjusted on age, sex, coinfections and symptoms delays.**

|  |  | Univariate |  |  |  | Multivariate |  |  |  |
|---|---|---|---|---|---|---|---|---|---|
|  | N | OR | 2.5% | 97.5% | p.val | AOR | 2.5% | 97.5% | p.val |
| All pathogens |  | 1.04 | 1 | 1.09 | 0.05 | 1.04 | 0.99 | 1.09 | 0.11 |
| Influenza A | 89 | 1.03 | 0.95 | 1.12 | 0.46 | 1.02 | 0.93 | 1.12 | 0.61 |
| SARS-CoV-2 | 74 | 0.99 | 0.91 | 1.08 | 0.81 | 1 | 0.9 | 1.1 | 0.99 |
| Rhinovirus/enterovirus | 92 | 1 | 0.9 | 1.11 | 1 | 1.02 | 0.92 | 1.14 | 0.65 |
| RSV | 39 | 1.07 | 0.9 | 1.31 | 0.41 | 1.049 | 0.83 | 1.41 | 0.69 |

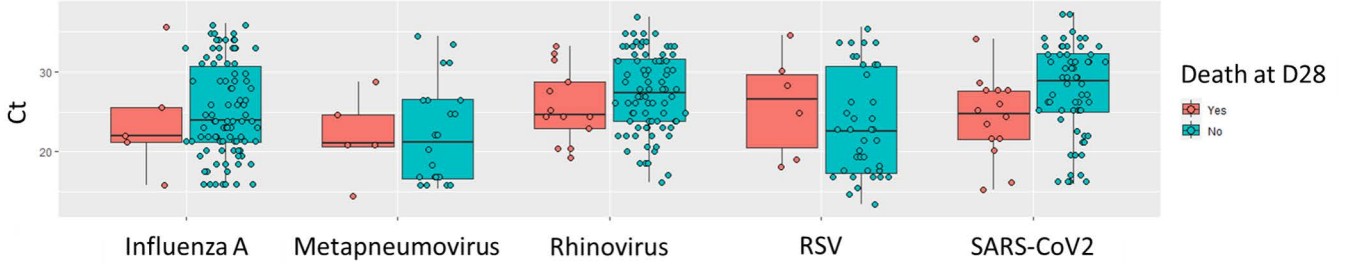

**Fig 2. Association between PCR cycle threshold (Ct) values and 28-day mortality.**

**Table 4. Association between PCR cycle threshold (Ct) values and mortality at day 28: univariate and multivariate analysis adjusted on age. sex. coinfections. and symptoms delays.**

| | Univariate | | | | Multivariate | | | |
|---|---|---|---|---|---|---|---|---|
| | OR | 2.5% | 97.5% | p.val | AOR | 2.5% | 97.5% | p.val |
| All pathogens | 1 | 0.95 | 1.05 | 0.9 | 1.01 | 0.96 | 1.07 | 0.67 |
| Influenza A | 0.96 | 0.81 | 1.12 | 0.64 | 0.96 | 0.8 | 1.12 | 0.59 |
| SARS-CoV-2 | 0.9 | 0.81 | 1 | 0.05 | 0.87 | 0.76 | 0.99 | **0.04** |
| Rhinovirus/enterovirus | 0.93 | 0.82 | 1.05 | 0.3 | 0.96 | 0.83 | 1.11 | 0.62 |
| Metapneumovirus | 0.99 | 0.83 | 1.17 | 0.91 | 1.03 | 0.8 | 1.32 | 0.78 |
| RSV | 1.05 | 0.92 | 1.22 | 0.47 | 1.12 | 0.95 | 1.36 | 0.19 |

## Discussion

This study assessed the association between Ct values and hospital outcomes in adult patients visiting the ED with suspected LRTI and tested with point-of-care multiplex PCR. While previous studies have reported an association between SARS-CoV-2 Ct values and hospital outcomes [13], limited data is available for other respiratory viruses and conflicting data have been reported for some viruses, notably influenza. Higher Ct values for SARS-CoV-2 tend to be associated with shorter hospital stays, as demonstrated by Hu and al. in a Chinese cohort in which the lowest Ct values were associated with a shorter LOS (OR = 0.707. 95% CI 0.680–0.735. P < 0.001) [14]. Several studies have confirmed this observation, including a study conducted in the United Kingdom in 2020 that reported an inverse association between high Ct values and death (aOR = 0.95. 95% CI 0.92–0.98; P = 0.001] [8] and a US study that reported an association between low Ct values and mortality (aOR = 6.05; 95% CI]. 2.92–12.52) and intubation (aOR = 2.73; 95% CI. 1.68–4.44) [15].

Surprisingly, despite the association between Ct values and mortality, we did not observe an association between SARS-CoV-2 Ct values and ICU admission, contrary to previous works such as Magleby et al. [15]. This might be related to the limited availability of ICU beds during the study period, during which patients that were eligible for ICU admission sometimes had to be hospitalized in the ED or in other wards.

Regarding influenza, the analysis did not objectify any association between Ct values and ICU admission or mortality, in accordance with several previous studies. This could be due to differences in physiopathology between influenza viruses and SARS-CoV-2 [16–19]. However, an association between higher Ct values for influenza, i.e., low viral loads, and prolonged length of stay was found. This finding, in apparent contradiction with the lack of association with severity, has previously been described in children by Fuller et al. [20]. This could be due to from several factors, such as a prolonged shedding of influenza viruses at a low level in patients with a prolonged length of stay such patients underlying comorbidities and/or poor health conditions, a role of bacterial surinfection in these patients or the importance of host response over viral replication in disease severity [21].

The analysis did not highlight any statistically significant association between Ct values and hospital length of stay, ICU admission or mortality for RSV. This might reflect a poor role of viral replication in the disease course or be due to the small number of RSV-positive patients. Indeed, such associations were reported in the pediatric population, including in a US study which enrolled 2615 children that reported that low Ct values were associated with more intensive care admissions (20%, compared with 15% and 16% in the groups with high and intermediate Ct values, respectively) and that there was a significantly increased risk of having a length of stay of ≥ 3 days in the groups with intermediate (odds ratio [OR], 1.43; 95% CI, 1.20–1.69) and high (OR, 1.58; 95% CI, 1.29–1.94) genomic loads [22].

No significant association between Ct values for HRV/EV and hospital outcomes were observed, while a few other studies had demonstrated significant associations between low Ct values and symptom severity or hospital LOS [15,18]. The large diversity of rhinoviruses and enteroviruses might explain the differences between studies for this family of

viruses. Rhinoviruses are effectively divided into over 160 serotypes in three species (A, B and C), with species C often being associated with more severe respiratory disease. Similarly, enteroviruses exhibit high genetic variability and can cause a wide range of clinical presentations, from mild respiratory symptoms to severe systemic infections. Furthermore, co-infection with other pathogens is frequent and complicates the clinical picture, potentially exacerbating symptoms or altering disease course. In some patients, a prolonged shedding of rhinoviruses and/or enteroviruses or rapid re-infections can be observed [23,24]. These factors are likely to explain the inconsistencies between studies investigating the relationship between these viruses and clinical outcomes, as highlighted in previous research) [25,26].

The QIAstat-Dx platform differs from point-of-care (POC) respiratory platforms such as the FilmArray and Cepheid GeneXpert systems in that it accepts a dry swab instead of a transport medium, simplifying sample collection and preparation. This feature, combined with its broad pathogen panel, makes it particularly suitable for use in Emergency Departments (EDs) where rapid and straightforward diagnostics are required. Ct values are available with different instruments but values can differ between platforms, as they are influenced by sampling, extraction and amplification protocols. This may limit the generalizability of results. Cross-platform comparative studies are needed to validate the diagnostic and prognostic utility of Ct values in different clinical contexts [27–29].

Our study is a retrospective monocentric study. Despite this limitation, its strength relies on the inclusion of a large number of adult patients that were tested at admission in the ED, using a point-of-care multiplex PCR and with homogenous indications throughout the study period. The study period allowed us to measure the association between Ct values and disease course for multiple respiratory viruses, including SARS-CoV-2. It should be noted that this study included the first waves of the COVID-19 pandemic. Thus, patients were not vaccinated but they could have mounted an immunity to previous SARS-CoV-2 infections. As SARS-CoV-2 sequencing was not performed, we could not measure the potential association between Ct values and hospital outcomes for specific variants. Additionally, the study focused exclusively on key clinical outcomes—hospital length of stay, ICU admission, and 28-day mortality—as these were predefined objectives. The analysis of respiratory symptoms was not planned as part of the original project and is therefore not included, limiting the scope of the findings regarding symptom-specific correlations.

## Conclusion

In conclusion, Ct values might help to assess disease severity and, thus, to predict hospital length of stay or mortality for influenza and SARS-CoV-2. However, this study was unable to pinpoint the link between Ct values and clinical outcomes for other respiratory viruses. Ct values can also be used to decipher if the patient is presenting with an acute infection or a prolonged viral shedding but confounding factors should be carefully considered (immunosuppression status, bacterial coinfection, duration of symptoms). To improve disease severity prediction, the use of Ct values in combination with markers of host response to viral infections and/or other biomarkers should be evaluated.

## Supporting information

**S1 Table. Symptoms, clinical patterns and outcomes according to the pathogen.**
(DOCX)

## Author contributions

**Conceptualization:** Donia Bouzid.

**Formal analysis:** Paul Loubet, Florian Salipante.

**Investigation:** Ines BENTAHAR, Christophe Choquet.

**Methodology:** Paul Loubet, Florian Salipante.

**Supervision:** Paul Loubet, Quentin Le Hingrat, Donia Bouzid.

**Validation:** Quentin Le Hingrat, Donia Bouzid.

**Writing – original draft:** Ines BENTAHAR, Paul Loubet, Donia Bouzid.

**Writing – review & editing:** Diane Descamps, Benoit Visseaux, Nathan Peiffer Smadja, Donia Bouzid.

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
