## [Decision Letter · Decision Letter 0]

14 Jan 2025

PONE-D-24-46715Respiratory viruses Ct values and association with clinical outcomes among adults visiting the ED with lower respiratory tract infections.PLOS ONE

Dear Dr. Bouzid,

Thank you for submitting your manuscript to PLOS ONE. After careful consideration, we feel that it has merit but does not fully meet PLOS ONE’s publication criteria as it currently stands. Therefore, we invite you to submit a revised version of the manuscript that addresses the points raised during the review process.

Editor's comments: 

1. The authors should discuss (1) similarity and difference of QIAstat platform and other POC respiratory platforms and (2) whether the findings about the Ct values is specific to QIAstat platform or can be generalized to other respiratory platforms. More references are needed, with the following reference as an example (citing is optional). 

Lakshmanan K, Liu BM. Impact of Point-of-Care Testing on Diagnosis, Treatment, and Surveillance of Vaccine-Preventable Viral Infections. Diagnostics. 2025;15(2):123. https://doi.org/10.3390/diagnostics15020123

2. Line 245-246: "The large diversity of rhinoviruses and enteroviruses might explain the differences between studies for this family of viruses." More references are needed, with this reference (PMID: 28031445) as an example (citing is optional).

3. 254-256 "As SARS-CoV-2 sequencing was not performed, we could not measure the potential association between Ct values and hospital outcomes for specific variants." This explanation does not make sense. SARS-CoV-2 sequencing usually demand clinical samples to have low Ct values. In other words, samples that can be successfully sequenced may not be suitable for observation for the potential association between Ct values and hospital outcomes for specific variants. The authors should discuss these points. More references are needed, with this reference (PMID: 39744807) as an example (citing is optional).

We look forward to receiving your revised manuscript.

Kind regards,

Benjamin M. Liu, MBBS, PhD, D(ABMM), MB(ASCP)

Academic Editor

PLOS ONE

Journal Requirements:

“Qiagen”

“DB contributed to a funded symposium for Qiagen

PL contributed to a funded symposium for Astra Zeneca”

5. Please amend the manuscript submission data (via Edit Submission) to include author Dr. Christophe Choquet.

6. Please ensure that you refer to Figure 1 in your text as, if accepted, production will need this reference to link the reader to the figure.

7. We note you have included a table to which you do not refer in the text of your manuscript. Please ensure that you refer to Tables 1-4 in your text; if accepted, production will need this reference to link the reader to the Table.

Reviewers' comments:

Reviewer's Responses to Questions

**Comments to the Author**

1. Is the manuscript technically sound, and do the data support the conclusions?

Reviewer #1: Yes

Reviewer #2: Yes

Reviewer #3: Partly

2. Has the statistical analysis been performed appropriately and rigorously? 

Reviewer #1: Yes

Reviewer #2: Yes

Reviewer #3: No

3. Have the authors made all data underlying the findings in their manuscript fully available?

Reviewer #1: Yes

Reviewer #2: No

Reviewer #3: No

4. Is the manuscript presented in an intelligible fashion and written in standard English?

Reviewer #1: Yes

Reviewer #2: Yes

Reviewer #3: Yes

5. Review Comments to the Author

Reviewer #1: Comments to Authors:

Minor points

1- Rewrite the title of manuscript; do not use the abbreviations in the title.

2- In abstract, page2, 35line\ In methods, page5, 108,119&147line\ In results,

page9, 173line\ In discussion,page13,219line, 224line, 226line, 234line and

255line\ In conclusion, page14, 260line; you wrote (11 times We)! The rule

of manuscript writing is to avoid using (We). So you should delete (We) and

use academic scientific words such as (This study or The current study or

The present study).

3- In results; you wrote the title of all tables below the tables!!! That’s

incorrect. Kindly transfer the table titles’ above the tables in addition to that

you should write the meaning of each abbreviation inside the tables below

each table.

Kind regards

Reviewer #2: The research is interesting and its statistical analysis is good. However, there are some data points that need to be clarified

1. Add data on how many people were admitted to the ICU and comorbidity factors to Table 1.

2. For Table 3, add ICU admission data for each virus.

3. Were comorbidity factors considered as confounding factors (not just co-infection) when analyzing the relationship between CT value and hospital length of stay, ICU admission, and mortality? If so, please include this.

4. Patient data with adenovirus, coronavirus HKU1, coronavirus OC43, influenza B, and parainfluenza infections were not analyzed separately, explain this in the manuscript.

5. There are only 28 patients with metapneumovirus infection; can this be analyzed?

Reviewer #3: 1. The manuscript needs to be proofread for grammar and typo corrections.

2. The manuscript aimed to determine a correlation between clinical outcome and the Ct values of patients visiting at the ED unit. Although there is much data available at hand, it is not comprehensively presented nor analysed in the manuscript. Why are only 3 criteria (LOS, ICU and 28-day mortality) assessed? How about respiratory symptoms? If they have in fact been analysed, results were not presented in detailed to show their association.

3. Methods section

a. Please indicate more clearly the inclusion criteria, the number of patient samples used.

b. Briefly describe the procedures involved using the QIAstat-Dx.

c. Kindly indicate the type of respiratory samples used for detection of the virus(es) (nasopharyngeal, nasal, etc)

4. There is no description of Table 1. Data in Table 1 can be segregated into different tables to make it clearer. What is displayed in Table 1 should be explained in the text as well.

5. All the other tables and figures are not explained clearly either. No reference to them in the text

6. The manuscript has a big potential to be published following comprehensive analysis and improvement of its presentation and content, however in its present form, I cannot recommend it to be published.

6. PLOS authors have the option to publish the peer review history of their article (what does this mean? ). If published, this will include your full peer review and any attached files.

**Do you want your identity to be public for this peer review?** For information about this choice, including consent withdrawal, please see our Privacy Policy .

Reviewer #1: No

Reviewer #2: No

Reviewer #3: No

---

## [Author Response · Author response to Decision Letter 1]

14 Feb 2025

Responses to the editor and reviewers’ comments

Editor's comments:

Dear Dr Liu,

Thank you for the opportunity to revise and resubmit our manuscript. We greatly appreciate your insightful feedback and have carefully addressed each of the points raised. Below, we provide a detailed response to your comments and outline the changes we have made to the manuscript accordingly.

1. The authors should discuss (1) similarity and difference of QIAstat platform and other POC respiratory platforms and (2) whether the findings about the Ct values is specific to QIAstat platform or can be generalized to other respiratory platforms. More references are needed, with the following reference as an example (citing is optional).

Lakshmanan K, Liu BM. Impact of Point-of-Care Testing on Diagnosis, Treatment, and Surveillance of Vaccine-Preventable Viral Infections. Diagnostics. 2025;15(2):123. https://doi.org/10.3390/diagnostics15020123

We have expanded the Discussion section to include a comparative analysis of the QIAstat platform with other widely used point-of-care (POC) respiratory testing platforms such as FilmArray (BioFire Diagnostics) and Cepheid GeneXpert. We highlighted the unique feature of the QIAstat instrument, which performs PCR testing for respiratory pathogens using a dry swab directly inserted into the cartridge.

We also addressed the generalizability of our results to other platforms by discussing potential differences in Ct value thresholds, assay sensitivity, and the influence of platform-specific protocols on Ct values.

To strengthen this section, we have included additional references, such as Lakshmanan et al. (Diagnostics, 2025), which outlines the broader implications of POC testing for respiratory pathogens.

“The QIAstat-Dx platform differs from point-of-care (POC) respiratory platforms such as the FilmArray and Cepheid GeneXpert systems in that it accepts a dry swab instead of a transport medium, simplifying sample collection and preparation. This feature, combined with its broad pathogen panel, makes it particularly suitable for use in Emergency Departments (EDs) where rapid and straightforward diagnostics are required.

Ct values are available with different instruments but values can differ between platforms, as they are influenced by sampling, extraction and amplification protocols. This may limit the generalizability of results. Cross-platform comparative studies are needed to validate the diagnostic and prognostic utility of Ct values in different clinical contexts.”

Réf:

• Lakshmanan K, Liu BM. Impact of Point-of-Care Testing on Diagnosis, Treatment, and Surveillance of Vaccine-Preventable Viral Infections. Diagnostics. 2025;15(2):123. https://doi.org/10.3390/diagnostics15020123

• PMID: 38015833

• PMID: 36851685

2. Line 245-246: "The large diversity of rhinoviruses and enteroviruses might explain the differences between studies for this family of viruses." More references are needed, with this reference (PMID: 28031445) as an example (citing is optional).

We have added several references, including the suggested PMID: 28031445, to better support our statement on the diversity of rhinoviruses and enteroviruses and its impact on clinical outcomes and study heterogeneity. This addition clarifies how genetic diversity, serotypes, and co-infections contribute to the observed differences across studies.

“Rhinoviruses are divided into over 160 serotypes in three species (A, B and C), with species C often being associated with more severe respiratory disease. Similarly, enteroviruses exhibit high genetic variability and can cause a wide range of clinical presentations, from mild respiratory symptoms to severe systemic infections. Furthermore, co-infection with other pathogens is frequent and complicates the clinical picture, potentially exacerbating symptoms or altering disease course. In some patients, a prolonged shedding of rhinoviruses and/or enteroviruses or rapid re-infections can be observed (PMID: https://doi.org/10.1183/09031936.00172113). These factors are likely to explain the inconsistencies between studies investigating the relationship between these viruses and clinical outcomes, as highlighted in previous research (PMID: 28031445- PMID: 35248716).”

3. 254-256 "As SARS-CoV-2 sequencing was not performed, we could not measure the potential association between Ct values and hospital outcomes for specific variants." This explanation does not make sense. SARS-CoV-2 sequencing usually demand clinical samples to have low Ct values. In other words, samples that can be successfully sequenced may not be suitable for observation for the potential association between Ct values and hospital outcomes for specific variants. The authors should discuss these points. More references are needed, with this reference (PMID: 39744807) as an example (citing is optional).

We agree with the reviewer that sequencing will not be feasible for samples with low viral loads, which could complicate the measurement of this potential association. One option would have been be to determine this association in pre-Omicron samples (before November 2021) and in samples collected after Omicron became the dominant variant (January 2022), and assume that all patients whose collected after January 2022 were infected with Omicron.

Reviewers' comments:

Reviewer #1: Comments to Authors:

We sincerely thank Reviewer #1 for their detailed and constructive feedback, which has helped us refine and clarify our manuscript.

Minor points

1- Rewrite the title of manuscript; do not use the abbreviations in the title.

The title has been revised to: "Respiratory Virus Cycle Threshold Values and Clinical Outcomes in Adults Visiting the Emergency Department with Lower Respiratory Tract Infections."

2- In abstract, page2, 35line\ In methods, page5, 108,119&147line\ In results,

page9, 173line\ In discussion,page13,219line, 224line, 226line, 234line and

255line\ In conclusion, page14, 260line; you wrote (11 times We)! The rule

of manuscript writing is to avoid using (We). So you should delete (We) and

use academic scientific words such as (This study or The current study or

The present study).

All instances of "We" have been replaced with neutral academic terms (e.g., "This study," "The analysis," etc.).

3- In results; you wrote the title of all tables below the tables!!! That’s

incorrect. Kindly transfer the table titles’ above the tables in addition to that

you should write the meaning of each abbreviation inside the tables below

each table.

Kind regards

Titles have been moved above the tables, and all abbreviations are now defined in the table legends

Reviewer #2: The research is interesting and its statistical analysis is good. However, there are some data points that need to be clarified

We thank Reviewer #2 for their thoughtful comments and valuable suggestions, particularly regarding the data analysis, which have strengthened our study

1. Add data on how many people were admitted to the ICU and comorbidity factors to Table

We appreciate the reviewer’s suggestion regarding the inclusion of ICU admission data and comorbidity factors in Table 1. To maintain readability and avoid overcrowding the table, these data have been provided in the supplementary table, where detailed information on ICU admissions and comorbidities is presented. This approach ensures clarity while allowing for a comprehensive overview of the study population.

2. For Table 3, add ICU admission data for each virus.

Table was modified as requested.

3. Were comorbidity factors considered as confounding factors (not just co-infection) when analyzing the relationship between CT value and hospital length of stay, ICU admission, and mortality? If so, please include this.

We appreciate the reviewer’s suggestion regarding the inclusion of comorbidities as confounders. As noted in the manuscript, the statistical models were already adjusted for potential confounding factors, including age, sex, coinfections, and symptom duration. However, we acknowledge that the explicit mention of comorbidities was not included in the description. To address this, we have clarified in the Methods section that comorbidities (e.g., chronic respiratory diseases, diabetes, cardiovascular conditions, and immunosuppression) were considered during the analysis to ensure comprehensive adjustment. Additionally, this clarification has been reflected in the Results section where relevant. Lines 136 and 168-169.

4. Patient data with adenovirus, coronavirus HKU1, coronavirus OC43, influenza B, and parainfluenza infections were not analyzed separately, explain this in the manuscript.

5. There are only 28 patients with metapneumovirus infection; can this be analyzed?

An explanation for the exclusion of certain viruses, including adenovirus, HKU1, OC43, and metapneumovirus, has been provided, noting that these pathogens were excluded from the detailed analysis due to small sample sizes, which limited the statistical power to draw meaningful conclusions.

Reviewer #3:

We greatly appreciate Reviewer #3's thorough review and insightful recommendations, which have significantly improved the presentation and depth of our work.

1. The manuscript needs to be proofread for grammar and typo corrections.

The manuscript has been thoroughly reviewed and revised to correct any grammatical errors and typos. We have ensured that the text is clear, concise, and adheres to standard academic English.

2. The manuscript aimed to determine a correlation between clinical outcome and the Ct values of patients visiting at the ED unit. Although there is much data available at hand, it is not comprehensively presented nor analysed in the manuscript. Why are only 3 criteria (LOS, ICU and 28-day mortality) assessed? How about respiratory symptoms? If they have in fact been analysed, results were not presented in detailed to show their association.

We thank the reviewer for this observation. The primary objective of this study was to assess the correlation between cycle threshold (Ct) values and key clinical outcomes, specifically hospital length of stay (LOS), ICU admission, and 28-day mortality. These outcomes were chosen a priori as they are critical indicators of disease severity and resource utilization in the emergency department setting.

The analysis of respiratory symptoms was not part of the original project plan and, therefore, was not included in this manuscript. Additionally, symptom presence and duration are inherently subjective and prone to variability in patient reporting, making them less reliable as primary outcomes. While additional analyses on symptoms could provide further insights, our focus on these three objective clinical outcomes aligns with the study’s predefined objectives. We have clarified this limitation in the Discussion section to provide better transparency regarding the scope of the study.

“Additionally, the study focused exclusively on key clinical outcomes—hospital length of stay, ICU admission, and 28-day mortality—as these were predefined objectives. The analysis of respiratory symptoms was not planned as part of the original project and is therefore not included, limiting the scope of the findings regarding symptom-specific correlations” Lines 280-285

3. Methods section

a. Please indicate more clearly the inclusion criteria, the number of patient samples used.

We thank the reviewer for raising this point. The inclusion criteria and sample information are already clearly described in the Methods section. Specifically, the study included adult patients (≥18 years old) presenting to the emergency department with symptoms of lower respiratory tract infection (e.g., dyspnea, cough, expectoration) accompanied by at least one general symptom (e.g., fever, myalgia) and requiring oxygen therapy. A total of 410 patients with positive multiplex PCR results were analyzed. However, to further ensure clarity, we have slightly restructured the text in the Methods section to emphasize these criteria and sample details.

b. Briefly describe the procedures involved using the QIAstat-Dx.

c. Kindly indicate the type of respiratory samples used for detection of the virus(es) (nasopharyngeal, nasal, etc)

We have now detailed both the procedures and the type of respiratory samples that were used in the Methods section:

“A nasopharyngeal swab was collected and tested using the QIAstat-Dx Respiratory Panel V2 (Qiagen. Hilden. Germany).”

More precisely, a dry nasopharyngeal swab was collected from each patient, this swab was inserted into the QIAstat-Dx cartridge, and a multiplex PCR was performed, allowing the detection of 22 different viruses and bacteria. When one of these microbial targets is amplified, the Ct value is available to the clinician and it is also stored in the Laboratory Information Management System.

4. There is no description of Table 1. Data in Table 1 can be segregated into different tables to make it clearer. What is displayed in Table 1 should be explained in the text as well.

We thank the reviewer for highlighting the need for a more detailed description of Table 1. In response, we have revised the corresponding paragraph in the Results section to provide a clearer and more comprehensive explanation of the data presented in Table 1. The revised text now includes a detailed narrative of the baseline demographic and clinical characteristics, including physiological measurements, pathogen detection rates, and cycle threshold (Ct) values. This ensures that all key findings from Table 1 are thoroughly described and contextualized. While we considered segregating Table 1 into multiple tables, we believe that presenting all baseline data in a single table maintains coherence and facilitates a better understanding of the study population. The updated paragraph ensures that the table is fully integrated into the text for clarity and alignment with the manuscript's flow. We hope these revisions address the reviewer’s concern and improve the manuscript’s readability.

“Four hundred ten patients (36.3%) with a positive multiplex PCR were included in this analysis, with a median age of 68 years (IQR: 55–80). A slight majority were male (55.4%, n=227). The median duration of symptoms was 3 days (IQR: 1–7), and 308 patients (75.1%) required hospitalization for at least one day, with a median hospital length of stay of 5 days (IQR: 1–11).

Baseline physiological and clinical measurements revealed a median pulse oximetry of 94% (IQR: 89–97%), a median respiratory rate of 20 breaths per minute (IQR: 17–28), and a median heart rate of 96 beats per minute (IQR: 83–112). Blood pressure values were also consistent with the patient cohort's clinical profile, with a median systolic pressure of 134 mmHg (IQR: 117–150) and diastolic pressure of 78 mmHg (IQR: 68–87). Glasgow scores indicated preserved consciousness, with a median score of 15 across the cohort.

The most frequently detected pathogens were rhinoviruses/enteroviruses (26.3%, n=108), influenza A viruses (24.9%, n=102), and SARS-CoV-2 (21.2%, n=87). Co-infections were identified in 37 patients (9%) with two pathogens and 2 patients (0.5%) with three pathogens. Median Ct values varied by pathogen, with influenza A showing a median of 23 (IQR: 21–30), rhinoviruses/enteroviruses 26 (IQR: 23.5–31), and SARS-CoV-2 27 (IQR: 22.4–31). Less frequently detected pathogens, including metapneumovirus, parainfluenza, and RSV, had median Ct values ranging from 21 to 31, depending on the pathogen (Table 1).” Lines 161-178

5. All the other tables and figures are not explained clearly either. No reference to them in the text

All the tables are now clearly cited in the manuscript, and their content more described in the manuscript.

6. The manuscript has a big potential to be published following comprehensive analysis and improvement of its presentation and content, however in its present form, I cannot recommend it to be published.

We sincerely appreciate the reviewer's recognition of the potential of our manuscript and their candid feedback. We have carefully addressed all the points raised to ensure a more comprehensive analysis and improved presentation of our study. Specifically, we have clarified th

---

## [Editor Report · Decision Letter 1]

20 Feb 2025

Respiratory viruses Ct values and association with clinical outcomes among adults visiting the ED with lower respiratory tract infections.

PONE-D-24-46715R1

Dear Dr. Bouzid,

We’re pleased to inform you that your manuscript has been judged scientifically suitable for publication and will be formally accepted for publication once it meets all outstanding technical requirements.

Kind regards,

Benjamin M. Liu, MBBS, PhD, D(ABMM), MB(ASCP)

Academic Editor

PLOS ONE
---

## [Editor Report · Acceptance letter]

PONE-D-24-46715R1

PLOS ONE

Dear Dr. Bouzid,

I'm pleased to inform you that your manuscript has been deemed suitable for publication in PLOS ONE. Congratulations! Your manuscript is now being handed over to our production team.

Kind regards,

on behalf of

Dr. Benjamin M. Liu

Academic Editor

PLOS ONE